# APOGEE 2: multi-layer machine-learning model for the interpretable prediction of mitochondrial missense variants

Salvatore Daniele Bianco[1,2], Luca Parca [1,3], Francesco Petrizzelli[1], Tommaso Biagini[1], Agnese Giovannetti [4], Niccolò Liorni[1,2], Alessandro Napoli[1], Massimo Carella[5], Vincent Procaccio[6,7], Marie T. Lott [7], Shiping Zhang[7,8], Angelo Luigi Vescovi[9], Douglas C. Wallace[7,10], Viviana Caputo[2] & Tommaso Mazza [1] ✉

Mitochondrial dysfunction has pleiotropic effects and is frequently caused by mitochondrial DNA mutations. However, factors such as significant variability in clinical manifestations make interpreting the pathogenicity of variants in the mitochondrial genome challenging. Here, we present APOGEE 2, a mitochondrially-centered ensemble method designed to improve the accuracy of pathogenicity predictions for interpreting missense mitochondrial variants. Built on the joint consensus recommendations by the American College of Medical Genetics and Genomics/Association for Molecular Pathology, APOGEE 2 features an improved machine learning method and a curated training set for enhanced performance metrics. It offers region-wise assessments of genome fragility and mechanistic analyses of specific amino acids that cause perceptible long-range effects on protein structure. With clinical and research use in mind, APOGEE 2 scores and pathogenicity probabilities are precompiled and available in MitImpact. APOGEE 2's ability to address challenges in interpreting mitochondrial missense variants makes it an essential tool in the field of mitochondrial genetics.

Mitochondria are responsible for many of the most important functions in eukaryotic cells. They use oxidative phosphorylation (OXPHOS) to produce large amounts of adenosine triphosphate (ATP), store calcium for cell signaling, generate heat, and mediate cell growth and death. Mitochondria, in contrast to the nuclear genome (nDNA), have a smaller repertoire of DNA-repair pathways. They counteract *Muller's ratchet*[1], the progressive accumulation of deleterious

mutations, with a process known as the *mtDNA bottleneck*, which, while controversial[2], explains the increase in cell-to-cell variability in terms of mutant load during development. A unique mechanism for the degradation of mutated DNA molecules[3] eliminates defective cells, allowing the mutant load to be stabilized over generations. Despite this, the mtDNA to nDNA variant ratios range from a few folds in non-vertebrates up to at least 20 folds in vertebrates[4]. The lack of

[1]Bioinformatics Laboratory, Fondazione IRCCS Casa Sollievo della Sofferenza, S. Giovanni Rotondo (FG), Italy. [2]Department of Experimental Medicine, Sapienza University of Rome, Rome, Italy. [3]Italian Space Agency, Rome, Italy. [4]Clinical Genomics Laboratory, Fondazione IRCCS Casa Sollievo della Sofferenza, S. Giovanni Rotondo (FG), Italy. [5]Medical Genetics Laboratory, Fondazione IRCCS Casa Sollievo della Sofferenza, S. Giovanni Rotondo, (FG), Italy. [6]University of Angers, Genetics Department CHU Angers, Mitolab UMR CNRS 6015-INSERM U1083, F-49000 Angers, France. [7]Center for Mitochondrial and Epigenomic Medicine, The Children's Hospital of Philadelphia, Philadelphia, PA, USA. [8]Department of Biomedical and Health Informatics, The Children's Hospital of Philadelphia, Philadelphia, PA, USA. [9]ISBReMIT Institute for Stem Cell Biology, Regenerative Medicine and Innovative Therapies, Fondazione IRCSS Casa Sollievo della Sofferenza, S. Giovanni Rotondo (FG), Italy. [10]Department of Pediatrics, Division of Human Genetics, The Perelman School of Medicine, University of Pennsylvania, Philadelphia, PA, USA. ✉e-mail: t.mazza@css-mendel.it

protective histones in mitochondria, the proximity of mtDNA to the electron transport chain, which is a primary cellular source of reactive oxygen species, or a dNTP pool imbalance that leads to decreased DNA polymerase gamma fidelity[5] are all factors that could affect these ratios.

Mutations in the mtDNA are at the core of many human diseases[6]. Currently, ~1000 different mutations are associated, based on literature, with human diseases in MITOMAP[7], ~10% of which with compelling evidence of being pathogenic based on published literature (named as *confirmed* in MITOMAP), ~86% have not been definitively shown to cause disease (named as *reported*), and 4% including synergistic and conflicting variants. Totally, 94% of confirmed and reported variants are single nucleotide variants (SNVs) and span the whole genome. Missense variants account for 43% of all SNVs in MITOMAP (Fig. 1a); 58% of them are present in the Genome Aggregation Database (gnomAD)[8] and 65% in HelixMTdb[9] (Fig. 1b).

As one might anticipate, the majority of gnomAD's missense variants are benign based on ClinVar, but gnomAD also includes twelve MITOMAP-confirmed missense variants and an additional three that ClinVar classifies as pathogenic (Supplementary Data 1). In the case of new variants or without a confirmed functional effect, one should apply the full 2020 ClinGen/ACMG/AMP guidelines[10]. For allele frequency (AF), they allow an evidence weight of "supporting" when a variant is found at an AF < 1/50,000 (<0.002%) in mitochondrial-specific databases (pathogenic criteria code "PM2"), "likely benign"

when AF > 0.5% ("BS1"), or "stand-alone benign" when AF > 1% and there is no other conflicting evidence, such as a novel occurrence in a major haplogroup branch to support pathogenicity ("BA1"). While confirmed variants are mostly rare[8], reported, conflicting, and synergistic variants are far more common. Indeed, there are eight confirmed variants in gnomAD with AF ≥ 0.002% (Supplementary Data 1) and 187 reported variants with AF ≥ 0.002%, of which 28 with AF ≥ 0.5% and 16 ≥ 1%. Similarly, HelixMTdb contains seven confirmed variants with frequencies ≥0.002% (Supplementary Data 1) and 191 reported variants ≥0.002%, of which 23 have AF ≥ 0.5% and 13 ≥ 1% (Fig. 1c). Because neither of the databases is enriched for mitochondrial disorders or other clinical phenotypes, compared to the GenBank mtDNA sequence repository, these variants are unlikely to be all pathogenic. As a result, variant pathogenicity can be challenging to predict as there are several biological mechanisms that concur with the functional behavior, e.g., epistasis and modulatory effects, which cannot be solely based on allelic frequency.

Inconsistencies were mitigated with the introduction of in silico prediction methods in 2015 by the ACMG and AMP's joint consensus recommendations[11], which were later modified[10] for the mitochondrial genome specifically. Variants were further given supporting evidence of benign status ("BP4") when multiple lines of computational evidence suggested no impact on the gene or gene product (coding nonsynonymous variants: APOGEE score ≤0.5; tRNA variants: MitoTip[12] <50th percentile and HmtVar[13] < 0.35). On the contrary, variants were

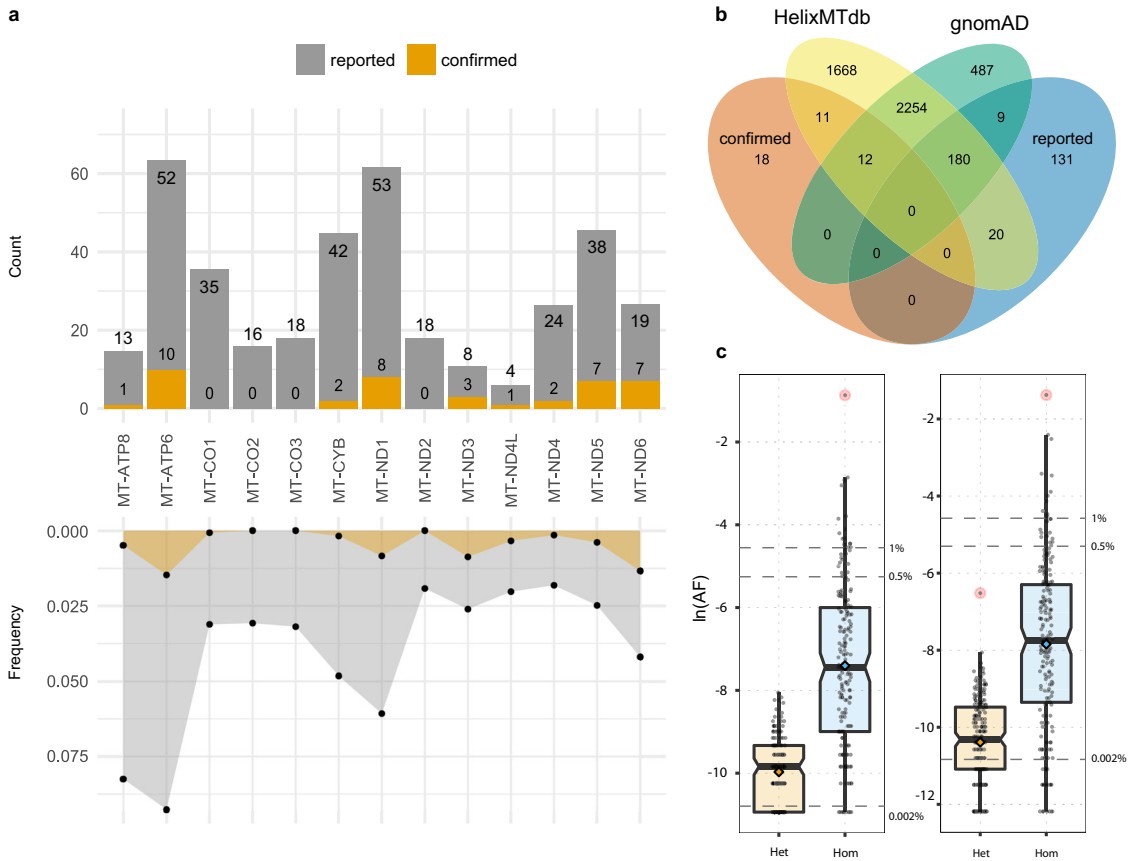

**Fig. 1 | Distribution of pathogenic and likely pathogenic missense variants in the mitochondrial genes and population databases. a** Counts (top) of reported and confirmed missense variants for all mtDNA protein-coding genes and their frequency (bottom) normalized on gene length. **b** Common missense variants between HelixMTdb, gnomAD, and MITOMAP's confirmed and reported variants. **c** Distribution of heteroplasmic (gnomAD, *n* = 164, HelixMTdb, *n* = 204) and homoplasmic (gnomAD, *n* = 187, HelixMTdb, *n* = 198) reported variants in gnomAD

(left) and HelixMTdb (right) based on their AF. Dashed lines represent the 0.002%, 0.5%, and 1% AF thresholds. Whiskers represent the 95% CIs around the median; the box limits represent the 25th and 75th percentiles (Q1 and Q3). GnomAD variants' AF values range from 1.77E−05 to 3.70E−04 (heteroplasmic) and from 1.77E−05 to 0.99 (homoplasmic). HelixMTdb variants' AF values range from 5.10E−06 to 1.47E−03 (heteroplasmic) and from 5.10E−06 to 0.99 (homoplasmic). Red dots are outlier variants by AF.

assigned an evidence weight of supporting pathogenicity ("PP3") when APOGEE > 0.5, MitoTip >50th percentile, and HmtVar ≥0.35.

The first version of APOGEE[14] was chosen over a slew of other options, the majority of which were designed to work with nuclear-encoded genes. They had poor prediction records when applied to mtDNA variants, evoking the historical congruency issue among predictors[15]. As a result, APOGEE was built as an ensemble method that was trained on mitochondrial-specific features that contributed to the best classification performance among all competitors.

Here, we present its latest iteration. With a better, state-of-the-art machine learning method and a curated training set, APOGEE version 2 improves its own performance. We tested its ability to spot pathogenic and neutral[16] variants in the mitochondrial genome and profiled its time-dependent "learning curve" to demonstrate how the steadily growing number of high-quality annotated mtDNA variants affects its classification performance. We have pre-calculated the prediction scores and pathogenicity probabilities of all possible missense variants of the mtDNA and set five classes of pathogenicity to support clinicians and geneticists in reporting their genetic diagnoses. Finally, we have conducted a mechanistic analysis of specific amino acids that cause perceptible long-range effects on the protein structure to discuss the significance of strengthening protein structural features in the training set.

## Results

### Characteristics of the training set

The APOGEE 2 training set (Dataset 1) contains 140 pathogenic and likely harmful variants and 1734 benign and likely harmless variants that have been manually curated (see Methods for a detailed description of the datasets). These were annotated with mitochondrially-tailored evolutionary, positional, and structural features and fourteen pathogenicity assessments of in silico predictors. Most of their Pearson correlation coefficients were in the range (−0.5, 0.5); MutationAssessor, SNAP, and PhD-SNP exhibited absolute Pearson's $r$ values > 0.7 with more than one other feature (Supplementary Fig. 1a and Supplementary Data 2).

Features were not always available for all variants. Missing values accounted for 1% of the evolutionary scores and in silico predictions and 0.2% of structural and positional values. We imputed them by using a Random Forest-based iterative imputer, which displayed the lowest normalized root mean squared error (NRMSE) values of all tested imputation methods (Supplementary Fig. 1b).

### Training, testing, and performance assessment

We examined different machine learning (ML) classification methods to sort deleterious variants, searching for the best-performing one, while all were designed to properly tackle the class imbalance in Dataset 1. Each method was instantiated during a 20-fold cross-validation (CV) repeated five times, where 19 folds of the dataset were iteratively used for the training and tuning of the hyperparameters, and the remaining fold was used for testing. Each method was tuned by attempting several combinations of hyperparameters (Supplementary Data 3). This was done using an inner 10-fold Grid-Search (GS) CV.

With the highest average test auPRC (0.716, Fig. 2a), auROC (0.95), and the best records for nearly all performance metrics (Supplementary Data 4), the KNN Bagging balanced through Random Under-Sampling (RUS) and Synthetic Minority Oversampling Technique (SMOTE) method (KNN_RusSmote) stood out as the best and was then chosen as the APOGEE 2 ML reference method. The best hyperparameters identified by GS were: "3" for the number of KNN's *neighbors*; the inverse Euclidean distance as a metric to weigh the neighbors' importance; 1/4 as the pathogenic/neutral variant ratio before the SMOTE step; and 1/4 as the ratio of features used by each base learner over the total number of selected features. It used 17 out of the 22 considered features, where PhastCons 100 V, MutationTaster,

FatHmmW, CADD, and ΔΔG were, in fact, discarded in the feature selection step (Fig. 2b).

On Dataset 1, APOGEE version 1 underwent a second round of testing and received an average auPRC of 0.573 (Fig. 2a) and auROC of 0.855. The classification performance of both APOGEE versions were also compared using the test set of our previous work[15], which included the union of MITOMAP and VariBench variants available at the time of that writing (see Supplementary Table 2 in[15]) purged of variants overlapping with the APOGEE 2 training. APOGEE 2 outperformed its former version (0.99 *vs.* 0.87 auROC, Fig. 2c; 0.97 *vs.* 0.65 auPRC).

Compared with other meta-predictors, APOGEE 2 exhibited the best performance metrics, including auPRC (Fig. 2d), but the sensitivity where MtoolBox and Condel excel at the expense of specificity, which ranked worst for both tools (Supplementary Data 4).

To evaluate whether expanding APOGEE 2's training set would improve its performance in future releases, we adopted the same criteria used to create Dataset 1 to generate seven training sets starting from the MITOMAP content from 2008 to 2020, which was randomly retrieved once every two years. The 2022 content was used as a test set. We found that the models' performance in sorting pathogenic from neutral variants in the 2022 dataset increased monotonically over time (Pearson's $r$: 0.91, $p$-value: 0.004) but at an increasingly slower rate (Fig. 2e, Supplementary Data 10), indicating that expanding the training set in subsequent iterations will not have as much of an impact on APOGEE 2's performance as adding more informative features would.

### Whole-genome predictions

We used APOGEE 2 to predict the pathogenicity of all 24,190 possible missense mtDNA variants (Supplementary Data 5) and made them available in MitImpact.

**Scores and pathogenicity probabilities.** Prediction scores are numeric and do not follow a conventional probability density function. The exact frequency distribution is shown in Fig. 3a. Scores range from 0 (neutral or benign) to 1 (deleterious or pathogenic). Using Bayesian reasoning, we could determine the posterior pathogenicity probability associated with all missense mitochondrial variants, ranging from 0 (not pathogenic) to 1 (pathogenic). Since the posterior probability monotonically increases with the APOGEE 2 score, ranking variants by one metric or the other is equivalent.

**Misclassification.** The misclassification analysis calculated for any of the 100 test folds revealed that 12.57% of the pathogenic variants were deemed benign, while 9.75% of the neutral variants were misclassified as deleterious (Fig. 3b).

**Positionality.** The dependency of the APOGEE 2 score on the variants' locations in the protein's 3D structures placed on a bisector of a 3D space (Fig. 3c) was quantified using Moran's index. For this analysis, we recalculated the APOGEE 2 scores by excluding the spatial features from the learning workflow in order to avoid any positional bias. We found significant positive spatial autocorrelation among the predicted pathogenic variants in each mitochondrial complex (one-sided Moran's index permutation $p$-value < 10E−5 for each complex except MT-ATP8, for which $p$-value = 0.02), implying that some regions of these proteins may be less tolerant to amino acid changes than others. We performed this analysis with the quadratic distance decay function as a measure of the proximity between amino acids. We also repeated the test using binary proximity matrices, computed at different cutoff distances as described in[17]. The maximum Moran's index has been obtained when setting a cutoff distance between amino acids of 6 Å for Complex I, Complex IV, and MT-ATP6, and 8 Å for Complex III (Fig. 3d); no significant results were obtained for MT-ATP8.

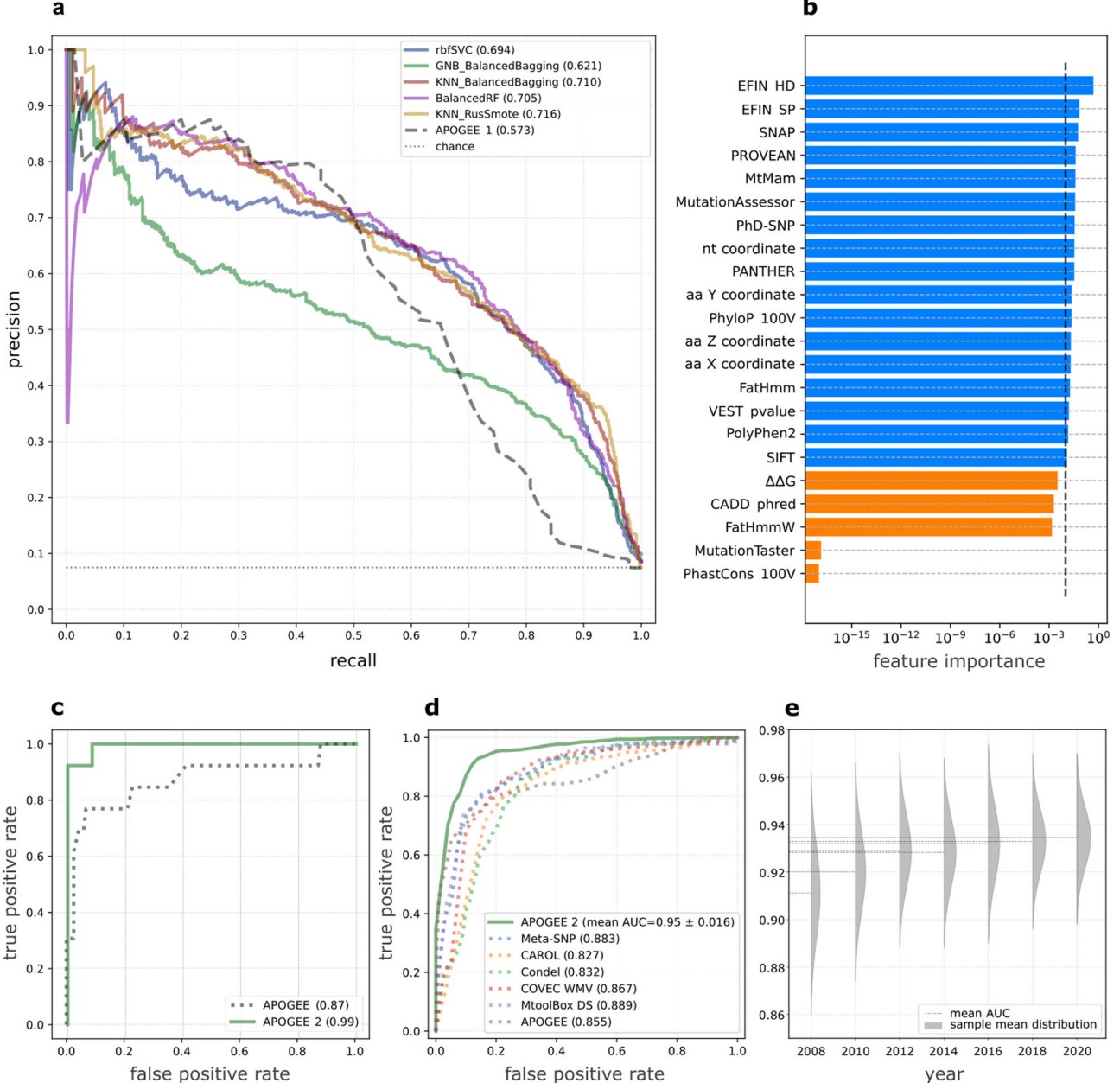

**Fig. 2 | APOGEE 2 performance evaluation. a** Average test auPRC values of the selected ML methods, calculated during the training phase. Support Vector Machine classifier with radial basis functions kernel (rbfSVC), Balanced Bagging using Gaussian Naive Bayes (GNB_BalancedBagging) and K-Nearest Neighbors (KNN_BalancedBagging) as base estimators, Balanced Random Forest (BalancedRF), KNN Bagging balanced through RUS and SMOTE techniques (KNN_RusSmote). **b** Feature importance assessed on the whole Dataset 1; threshold set to 1%. **c** AuROC values calculated on 118 and 13 neutral and pathogenic test variants for APOGEE versions 1 and 2. **d** Performance comparison of APOGEE 2 *versus* other meta-predictors in terms of auROC. APOGEE 2's auROC is reported as the mean ±95% CIs obtained through cross-validation. **e** Time-dependent APOGEE 2's auROC values obtained by predicting MITOMAP 2022 upon training on the 2008–2020 contents; for each year, the sample mean distribution is reported in gray.

We localized spatially autocorrelated high- and low-risk regions using the LISA index. The most significant regions (adjusted one-sided LISA permutation *p*-value < 0.01) were reported in Supplementary Data 6. The APOGEE 2 score threshold used to sort low-from high-risk amino acids in this analytical step was 0.379, which is the median recalculated APOGEE 2 score. For example, focusing on Complex I, which is composed of seven mtDNA-encoded proteins, we identified several neutral hotspots spread throughout MT-ND2 and several fragile regions, which included the transmembrane helices of MT-ND5, MT-ND4, and MT-ND1, the transmembrane helix 3 (TMH3) of MT-ND6 and the MT-ND3 loop. The latter two regions include variants known to impact important physiological

mechanisms, which were examined more deeply in the "Evaluation of variants that alter the protein structure non-locally" section (Fig. 3e).

**Categorization.** Interpreting the evidence categories given in Tables 3 and 4 of[11] as categorical, conditional probabilities or odds of pathogenicity, in line with[18], and aiming to provide clinical translationality to APOGEE 2 predictions, we adopted the following ranges of probability and set four classes of pathogenicity: benign ≤0.001, 0.001 < likely benign ≤ 0.1, 0.9 ≤ likely pathogenic < 0.99, pathogenic ≥ 0.99. When none of the previous criteria are met, i.e., when the posterior pathogenicity probability of a variant is between 0.1 and 0.9, a variant has

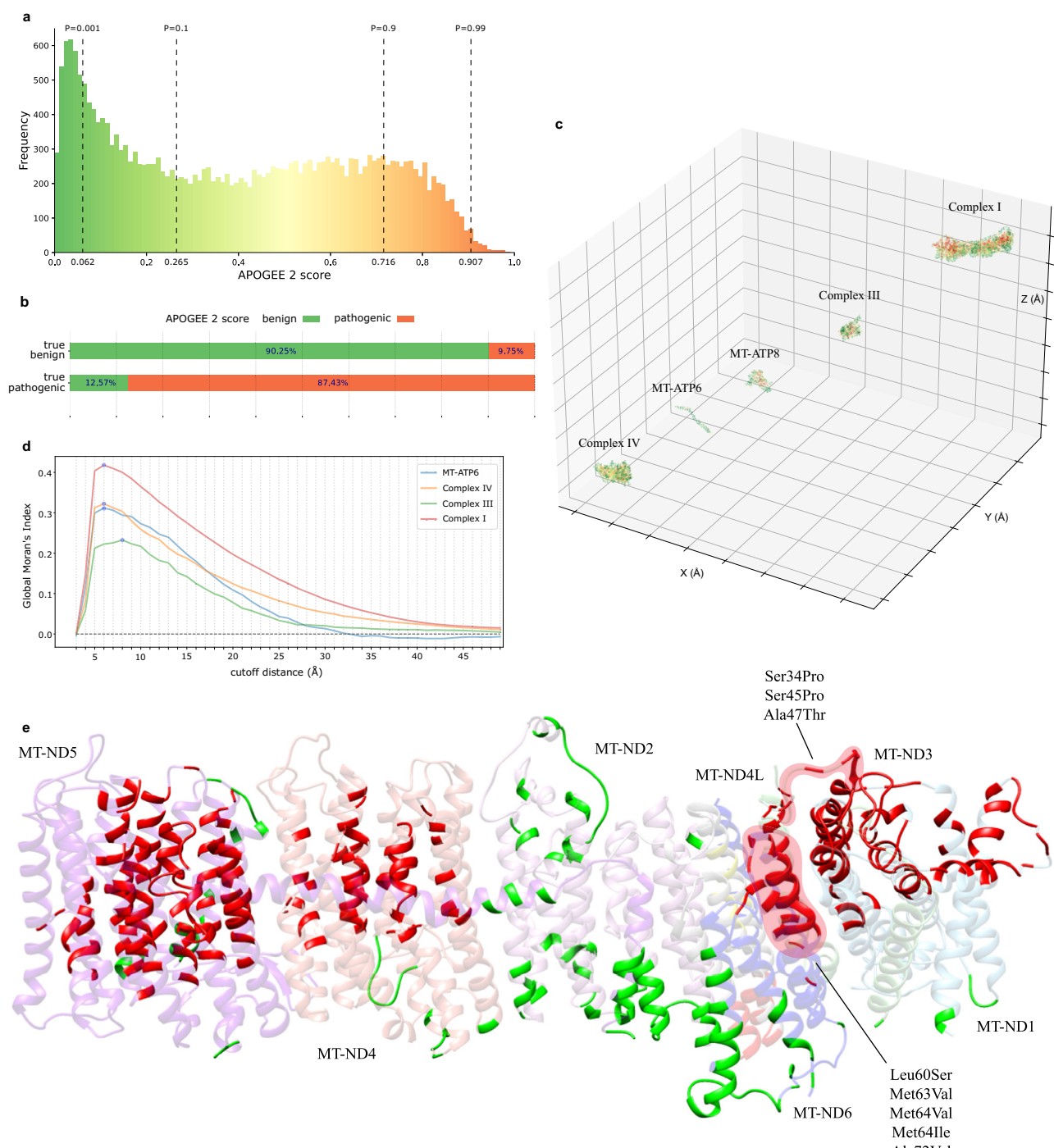

**Fig. 3 | APOGEE 2 scores distribution and spatial autocorrelation. a** Distribution of APOGEE 2 scores. Colors represent classes of pathogenicity: green (benign, probability of pathogenicity (P) ≤ 0.001, score (S) ≤ 0.062), light green (likely benign, 0.001 < P ≤ 0.1, 0.062 < S ≤ 0.265), yellow (VUS, 0.1 < P < 0.9, 0.265 ≤ S < 0.716), orange (likely pathogenic, 0.9 ≤ P < 0.99, 0.716 ≤ S < 0.907), red (pathogenic, P ≥ 0.99, S ≥ 0.907). **b** Misclassification rate of 100 test folds calculated on Dataset 1. **c** Mitochondrial protein complexes localization on the bisector of a 3D space. Colors have the same meaning as Fig. 3a. **d** Global spatial auto-correlation computed at different cutoff distances. Blue circles mark the maximum values for each protein complex. **e** Low-risk (green) and high-risk (in red) amino acid regions of the mitochondrial Complex I subunits. Highlighted in red, we underline the MITOMAP confirmed variants that localize on TMH3 of MT-ND6 and on the MT-ND3 loop.

uncertain significance (VUS) (Supplementary Data 5). With this setting, all known pathogenic variants in Dataset 1 were correctly labeled pathogenic and likely-pathogenic, except for seven of them, which were annotated as VUS. Regarding the harmless variants in Dataset 1, 190 were annotated as VUS, while the remaining 1544 were correctly labeled benign and likely-benign (Supplementary Data 5). Nearly all variants in Datasets 2 and 3 (Supplementary Data 7), which contain

only potentially harmless variants and therefore were used to evaluate APOGEE 2's specificity, were classified as benign or likely-benign by APOGEE 2 ($\chi^2(1, N = 36) = 16.19$, *p*-value = 5.73E-05 and $\chi^2(1, N = 35) = 24.10$, *p*-value = 9.15E−07, respectively for Datasets 2 and 3). None of the variants in Dataset 2 were classified as pathogenic or likely-pathogenic, while one variant from Dataset 3 was classified as likely-pathogenic.

## Variant functionality prediction in human variation databases

As a further test of the negative variants, we used APOGEE 2 to score every mitochondrial missense variant in the gnomAD v3.1.2 and HelixMTdb databases. Heteroplasmic variants were 1767 (AF range min 0.0017% - max 0.037%) in gnomAD and 3058 (AF range min 0.0005% - max 0.15%) in HelixMTdb. The homoplasmic variants were 2177 (AF range min 0.0017% - max 99%) and 2894 (AF range min 0.0005% - max 99%). The predicted pathogenic and likely pathogenic variants for both databases were comparable (Supplementary Fig. 2a, b), even when considering AF < 0.002% variants, irrespective of their heteroplasmy levels. GnomAD and HelixMTdb showed, in fact, a high and significant positive correlation between the allelic frequencies of their heteroplasmic (Pearson's $r$: 0.88, $p$-value < 2.2E−16) and homoplasmic (Pearson's $r$: 0.91, $p$-value = 6.434E−11) variants.

It is important to note that HelixMTdb was not enriched in patients with mitochondrial diseases, but there were no exclusion criteria based on mitochondrial disorders[9]. On the other hand, while gnomAD declares that some individuals with severe disease may still be included in the database, albeit likely at a frequency equivalent to or lower than that seen in the general population, they have explicitly removed individuals known to be affected by severe pediatric diseases, as well as their first-degree relatives. This might explain the presence of both likely and, frankly, pathogenic variants in both datasets.

The pathogenicity probability values for ClinGen's neutral variants, which were collected in Dataset 4 (Supplementary Data 7), ranged from 0 to 0.73. Dataset 4 (see Methods) contains neutral variants that were evaluated by the ClinGen mitochondrial VCEP team and, therefore, are supposedly more likely to be neutral than those contained in Datasets 2 and 3. None of Dataset 4 variants were predicted as pathogenic or likely-pathogenic by APOGEE 2. We also verified that the APOGEE 2 scores of these variants were lower on average than those of the neutral variants in Dataset 1 (one-sided Mann−Whitney $U$ test, $U = 152,940$, $p$-value = 2.80E−09; Supplementary Fig. 2c), thereby confirming the ability of APOGEE 2 to also quantify the level of certainty of being a neutral variant.

## Evaluation of variants that alter the protein structure non-locally

Variants that have structural and non-local effects may significantly impair APOGEE 2 prediction performance. This is the case of NC_012920.1:m.10161 A > C[19], (YP_003024033.1:p.Thr35Pro), which is located in the MT-ND3 loop (residues 24−54) and is contiguous to the m.10158 T > C (p.Ser34Pro) common variant, reported as "confirmed" by MITOMAP, as "pathogenic" in ClinGen, and is associated with Leigh disease or MELAS syndrome. The loop is between two transmembrane helices (TMH 1 and 2) and includes Cys39. In mammalian Complex I, such a residue is exposed during active mitochondrial respiration and is thought to be necessary for the reversible transition between catalytically active and inactive states[19]. Intuitively, the loop's dynamics may influence Cys39 exposure and, as a result, the active-inactive state transition. Multiple MD simulations revealed that the Ser34Pro and Thr35Pro mutants affect the loop flexibility significantly and similarly to the wild-type protein, and the other two mutant systems, Ser34Phe and Ser34Tyr (Fig. 4a−d), which were chosen because of their allelic frequencies (0.013% and 0.002% in MITOMAP, respectively), were contiguous and not associated with any relevant phenotype. The RMSF profiles (Fig. 4c) of the heavy atoms in part of the loop (residues 40−50) were higher for Ser34Pro and Thr35Pro than the wild-type and slightly more rigid in the first part (residues 24−40). On the other hand, Ser34Phe and Ser34Tyr displayed flexibility profiles that were similar to those of the wild-type. This can also be observed in the 3D dynamic representation (Fig. 4d). The loss of essential interactions between loop residues and nearby subunits brought on by Thr35Pro, such as residues 129 of MT-ND1 and 49 of MT-ND3 and residues 76 of MT-ND6 and 48 of MT-ND3, is what leads to the flexibility alterations. APOGEE 2

classifies this variant as VUS (score 0.51, probability = 0.59) even though we have confirmed it to be pathogenic (data not shown).

A second case regards NC_012920.1:m.14538 A > G, YP_003024037.1:p.Phe46Leu in MT-ND6, reported in the literature[20] as a novel pathogenic LHON variant but classified as likely-benign by APOGEE 2 (score 0.08, probability = 0.002). It is proximal to the transmembrane helix TMH3 (residues 52−74) in the closed state and interrupted by a bulge in the open state that involves residues 60−65[21] of MT-ND6, which actively participates in the Complex I closed conformation, rotating to lose its characteristic π-bulge and consequently disrupting the helix[22]. A few pathogenic mutations were found in the helix. One of them, m.14459 G > A, hits the residue Ala72, reported in MITOMAP and ClinGen as a pathogenic change in valine and associated with multiple mitochondrial phenotypes. It is interesting to note that the same residue is also hit by a proline variant (Ala72Pro, m.14460 C > G). The proline variant is included in Dataset 1 because it is deemed benign due to its allelic frequency of 0.011%, but it is categorized as VUS by APOGEE 2 (score = 0.64, probability = 0.82). We have compared the long-range impact of p.Phe46Leu on TMH3 with that caused by Ala72Pro and Ala72Val by simulation. After 200 ns of simulation, we observed that all three mutants altered the helix's folding to different extents, with a more evident destructuration caused by Ala72Pro and Ala72Val than Phe46Leu (Fig. 4e).

## Discussion

APOGEE 2 is an ensemble method that addresses the need for a pathogenicity prediction tool with increased accuracy for interpreting missense mtDNA variants. We have shown that it outperforms the former version due to the availability of more mitochondrially-centered features, their manual curation, and the adoption of a more sophisticated ML protocol. Overall, APOGEE 2 offers the best performance metrics when compared to other meta-predictors. It is interesting to note, though, that Condel and MtoolBox excel in sensitivity at the expense of their specificity (Supplementary Data 4). APOGEE 2 outperforms several other predictors as well, i.e., MutPred and MutPred2[23], and MutationAssessor, CADD, and EFIN (Supplementary Fig. 3), which were used for APOGEE 2's training.

EFIN HD is the only predictor with a higher specificity score. It identifies 1633 of 1734 likely neutral variants *versus* 1544, which were identified by APOGEE 2. It should be noted, though, that the degrees of certainty of the pathogenic and neutral variants of our training set differ. The pathogenicity of variants is based on the literature. Neutrality is assumed by allelic frequency considerations and available MITOMAP annotations. Given that some variants with literature reports of suspected pathogenicity show frequency levels higher than 0.002% in gnomAD and HelixMTdb, it is also likely that some variants that make up the neutral subset of Dataset 1 are actually pathogenic. The reverse is not obvious. We have then tested APOGEE 2 and EFIN HD on ClinGen's recently curated neutral variants; in general, APOGEE 2 assigned lower pathogenicity scores to ClinGen's neutral variants than EFIN HD did (one-sided Wilcoxon signed-rank test preceded by quantile transformation of both scoring systems; the EFIN score has been inverted since it is proportional to the neutrality of the variant; ranks sum = 3236, $p$-value = 0.003).

We have demonstrated that the APOGEE 2 pathogenic scores exhibit significant spatial autocorrelation, suggesting that some protein structural regions may be less tolerant to amino acid changes than others. As a result, APOGEE 2 can predict mitochondrial-specific high and low-risk regions, but it may still be ineffective for variants that have non-local structural effects, particularly if the involved regions are low-confidence/destructured. The flexibility of an MT-ND3 loop, for example, seems to be the key feature for evaluating the functionality of Thr35Pro. Thr35Pro and Ser34Pro were shown to change the flexibility of the loop-spanning residues 24−40 more than Ser34Tyr and Ser34-Phe, two rare variants with no associated phenotypes and dynamics that were similar to the wild-type protein. The motion of this loop was

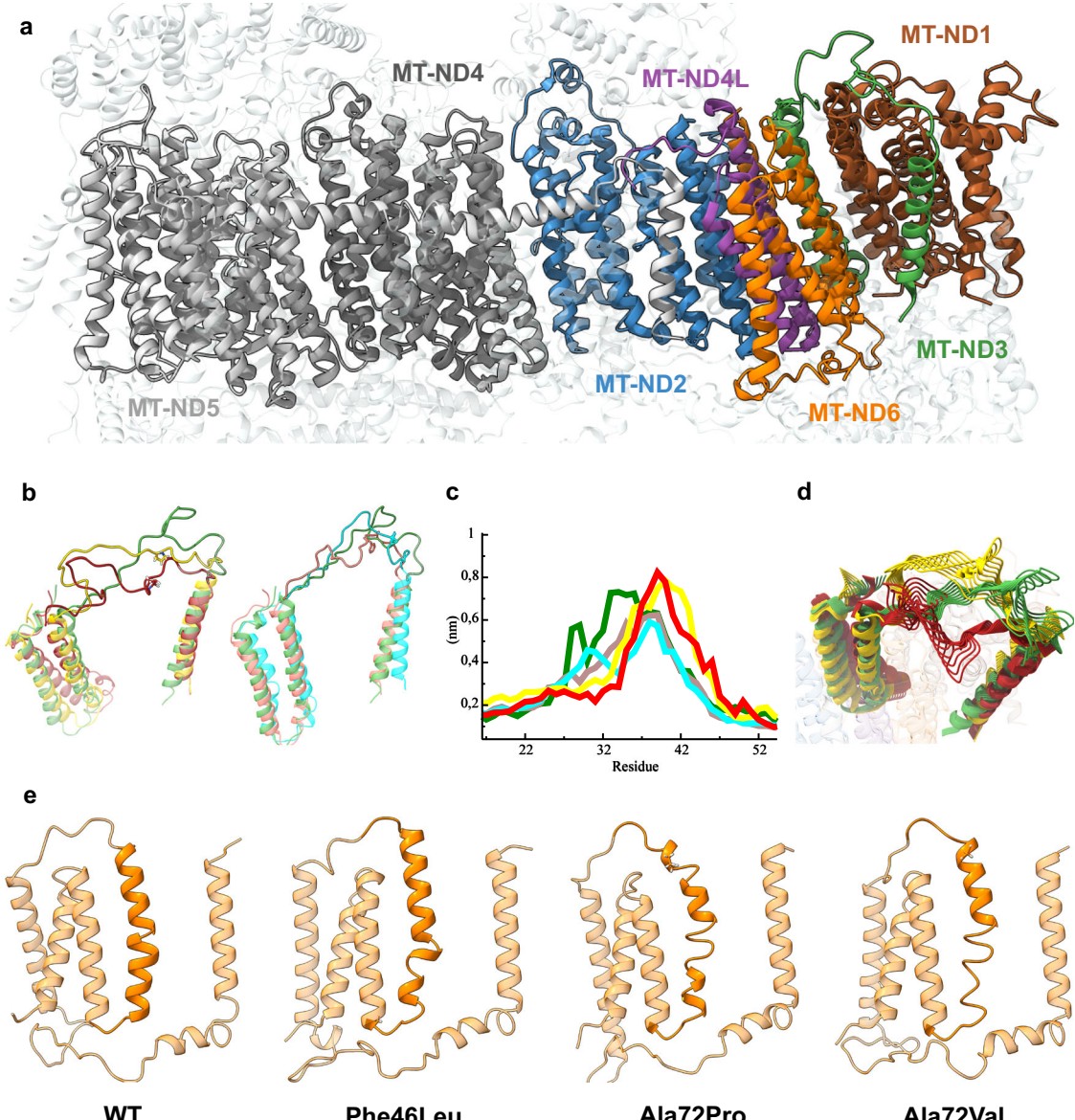

**Fig. 4 | Long-range effects analysis through molecular dynamics simulation.**
**a** Structure of the mtDNA-encoded subunits of the complex I membrane arm.
**b** Average structures of the wild-type, Ser34Pro, and Thr35Pro MT-ND3 protein
models (left) and wild-type, Ser34Phe, and Ser34Tyr (right). **c** RMSF profiles of the
heavy atoms of the MT-ND3 loop (residues 24–54) for both wild-type and mutants.

**d** 3D representations of the dynamics of the wild-type, Ser34Pro, and Thr35Pro MT-
ND3 protein models. In all subfigures **b**–**d**, wild-type is colored green, Ser34Pro is
yellow, Thr35Pro is red, Ser34Phe is pink, and Ser34Tyr is cyan. **e** Average structures
of the MT-ND6 protein. TMH3 is highlighted in dark orange.

critical in the establishment of essential links with neighboring sub-
units, which were broken by Thr35Pro. The m.14538 A > G, p.Phe46Leu
pathogenic variant is deemed likely-benign by APOGEE 2 because its
effect is not localized but instead works to disrupt the nearby TMH3
helix, which in turn plays a crucial role in the Complex I closed
conformation.

This factor may also have an impact on how missense variants
are interpreted in terms of their pathogenic potential when com-
bined with other variants, e.g., synergistic variants, or in the context
of particular mtDNA haplogroups[24]. Synergistic variants, in parti-
cular, can be challenging to assess as they can be common in control
populations and involve poorly conserved residues[25]. These con-
siderations may explain the fact that only one of the 17 missense
variants noted in their original publications as being possibly
synergistic[7] is predicted as likely-pathogenic by APOGEE 2, which
instead classified the remaining 4 as benign, 9 likely-benign, and 3
VUS (Supplementary Data 8).

Narrowing the number of VUSs and, therefore, performance
improvements are possible by adding more accurate structural fea-
tures to variants and accounting for their epistatic interactions, still
considering that the growth and curation level alone of the true
positive MITOMAP variants in the past 14 years contributed to a
smaller and smaller monotonic increase in the APOGEE 2 performance.
Therefore, thresholds of pathogenicity will undoubtedly change as
more features are added to the predictor, and more clinical or func-
tional evidence is published. Intra-VUS soft thresholds are desirable
since clinicians and researchers may be especially interested in dif-
ferentiating between a high-scoring VUS (i.e., closer to the likely
pathogenic threshold) and a low-scoring VUS (i.e., closer to the likely
benign threshold). They may want to investigate high or low-scoring
VUSs deeper with additional functional assays with the ultimate goal of
moving variants to the final category, i.e., likely pathogenic or likely
benign. An indication of this has been added to the MitImpact live
report. This will be an ongoing, dynamic process, and users are

encouraged to refer to the website for the most up-to-date thresholds and scores.

Molecular dynamics simulations of a significantly high number of variants that may, even if distantly, alter the functional structure of critical components of the mitochondrial proteins or cooperate with other variants to cause a significant respiratory-chain deficiency might contribute in this direction. A pilot project has already started, and preliminary results are available on MitImpact's website (https://mitimpact.css-mendel.it/).

To facilitate the use of the APOGEE 2 scoring and pathogenicity probabilities by clinicians and researchers, we have pre-computed these calculations for all missense variants and made them available in MitImpact through its web interface and as a flat file. Variant curators using the current ACMG/AMP modified guidelines for mtDNA variants[10] should preferentially use the APOGEE 2 scores. Bayesian probabilities of pathogenicity are also given for those using the APOGEE 2 predictions on a standalone basis.

## Methods

### Datasets

Four disjoint sets of variants were used in this work. Dataset 1 comprises 1874 non-synonymous mtDNA variants, divided into 1734 deemed benign and 140 pathogenic variants (Supplementary Data 5). The former set was obtained from MITOMAP's "general" variants (accessed on April 20, 2021), which were purged of overlapping MITOMAP's "disease" variants and variants with ClinGen's pathogenic criteria code "PM2." The pathogenic set was made up of MITOMAP "disease" variants, including 41 *confirmed* and 99 *reported* variants. Overlapping MITOMAP's "general" variants were discarded. Neither set contained any synergistic or conflicting variants. Dataset 2 was obtained from gnomAD version 3.1.2. It contained 36 missense variants not present in the training set, which exhibited allelic frequencies ≥0.002% both in heteroplasmy and homoplasmy. Dataset 3 was extracted from HelixMTdb ver. 20200327 and filtered as Dataset 2. Twelve variants were shared with Dataset 2 and discarded, and 35 were unique to this dataset. Dataset 4 fetches variants from the benign/likely-benign variant sets approved by the Mitochondrial Variant Curation Expert Panel (VCEP) of ClinGen (https://clinicalgenome.org/) as having met the following criteria for mtDNA variants[10]: benign variants in the set had allele frequencies >1% ("BA1") in either MITOMAP, gnomAD, or HelixMTdb and without disease reports or negative in silico predictors; likely benign variants in the set had allele frequencies of 0.5–1% ("BS1") in MITOMAP, gnomAD, or HelixMTdb and without disease reports or negative in silico predictors, and additionally had either a supporting in silico tool score for benignity or a resulting synonymous amino acid change. All other variants meeting the AF cutoffs but with possible disease associations are referred to the ClinGen mitochondrial VCEP for individual curation. The current set of benign/likely-benign variants meeting the ClinGen criteria was accessed for this study on September 7, 2022, and released while Dataset 1 was already made up. We gathered 135 neutral missense variants in total, dropped 8 VUS variants, and found that 97 of them overlapped Dataset 1. Datasets 2–4 were used to evaluate the specificity of the APOGEE 2 predictions (Supplementary Data 7).

### Features

APOGEE 2 was trained on three classes of information: evolution, pathogenicity predictions, and protein structural features.

### Evolution

PhyloP and PhastCons conservation scores on 100 vertebrate species were obtained from the UCSC Table Browser. Empirical substitution scores were obtained from the MtMam rate matrix, built on 12 proteins, i.e., ATP6, ATP8, COX1, COX2, COX3, CYBB, ND1, ND2, ND3, ND4, ND4L, and ND5, which are located on the heavy strand of the mtDNA (3331 sites)[26]. Data are from 20 species of mammals and three close outgroups, i.e., Wallaroo, Opossum, and Platypus. The rate matrix was downloaded from https://github.com/abacus-gene/paml/blob/master/dat/mtmam.dat.

### Pathogenicity predictors

Pathogenicity scores were retrieved for the following software packages: PolyPhen2[27], SIFT[28], fathmm[29], PROVEAN[30], MutationAssessor[31], EFIN[32], CADD[33], PANTHER[34], PhD-SNP[35], SNAP[36], and MutationTaster2[37].

### Structural features

The 3D structures of the 13 proteins were retrieved from the RCSB Protein Data Bank (PDB) with the following IDs: 5xtc for the Respiratory Complex I (s, i, j, r, k, l, and m chains for MT-ND1, MT-ND2, MT-ND3, MT-ND4, MT-ND4L, MT-ND5, and MT-ND6, respectively); 5z62 for the Cytochrome C Oxidase (chain A for MT-CO1, chain B for MT-CO2, and chain C for MT-CO3); 5xte for the Respiratory Complex III (chains J and V for Cytochrome b). Since MT-ATP6 and MT-ATP8 were not associated with any resolved X-ray structure, their predicted structures were downloaded from the AlphaFold2[38] Protein Structure Database[39]. The five PDB files were placed on the bisector of a 3D space and spaced $3d$ apart, where $d = 20$ Å. The total energy variation ($\Delta\Delta G$, $K_{cal}$/mol) eventually caused by any possible non-synonymous amino acid change was calculated as follows. The structures/models described above were first repaired (FoldX's module *RepairPDB*) and then mutated (*PositionScan*), considering all possible amino acid changes causing missense variations. Each mutant was analyzed energetically using FoldX ver. 5.0[40] in comparison with the native structure. FoldX was run with default parameters.

All these features, except for $\Delta\Delta G$ values, mtMam, and the spatial coordinates of amino acids, were pre-calculated and made available in MitImpact ver. 3.0.6[41].

### ML workflow

The overall learning procedure consists of a double cross-validation procedure that nests an optimization procedure of the best parameter combinations (hyperparameters) of each implemented ML algorithm under a model selection procedure. The workflow was implemented in Python and made available at https://github.com/mazzalab/playgrounds [51] as a Colab notebook.

In particular, for each ML algorithm, the whole dataset was primarily split into training and test sets by a stratified 20-fold cross-validation (CV) procedure, implemented in Python's *scikit-learn's StratifiedKFold*, and repeated five times. Then, for each unique group, we took it as a holdout and used the remaining groups (19) as a training data set. Each group was guaranteed to maintain the original pathogenic/neutral variant ratio. Then, each training set was progressively subjected to three preprocessing steps: *scaling*, *imputation* of missing values, and *feature selection*.

### Data preprocessing

Values were scaled using the Python *scikit-learn's StandardScaler* module to zero mean and unit variance. The scaler was trained on the training-set features and then used to transform both the training and test sets' values (Fig. 5a). Pairwise correlations between the scaled values to zero mean and unit variance of non-positional features, namely those not directly involving the genomic or amino acidic positions of variants, for all possible mitochondrial missense variants were obtained using Pearson correlation coefficient. Then, we imputed missing values (Fig. 5b), as explained in the section below. Finally, a supervised feature selection step was implemented using a Decision Tree classifier (*DecisionTreeClassifier scikit-learn* module) to assess the relative information gain associated with a feature and hence its contribution to the overall decision process. The features that explained less than 1% of the total information gain were dropped. This step was performed on the training set only.

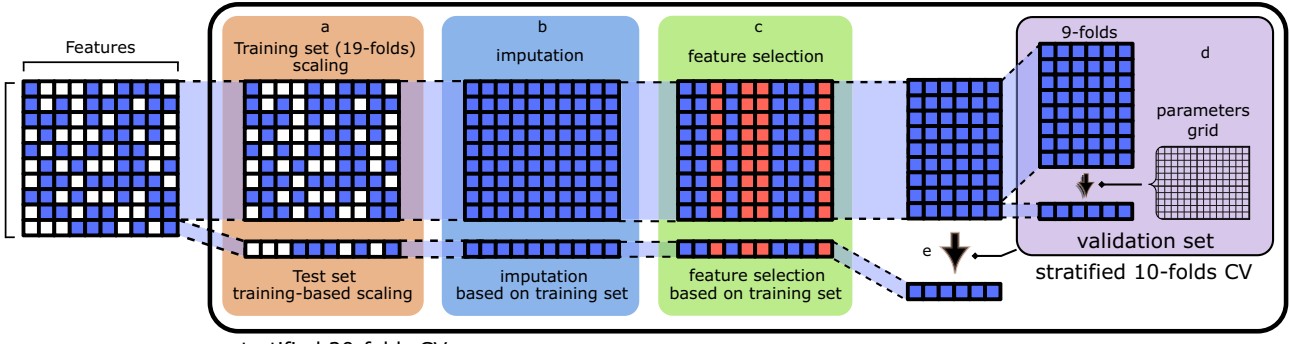

**Fig. 5 | APOGEE 2 ML pipeline.** It includes data preprocessing, i.e., scaling (**a**), imputation of missing values (**b**), and feature selection (**c**), model tuning by 10-folds Grid Search CV (**d**), training of an ML method with the best hyperparameter combination obtained in (**d**) and testing (**e**).

Then, the dropped features were removed from the training and test sets (Fig. 5c).

## Imputer selection

The whole MitImpact content, made up of 22,316 variants regardless of their label (neutral or pathogenic) and excluding the variants in Dataset 1, was split into 20 folds. Each fold was iteratively chosen as a test- and the remaining 19 as training variants for each of the following five imputation methods. We implemented two simple interpolation methods based on the mean and median values of the features to be imputed, respectively; a k-Nearest Neighbors imputer (sklearn.impute.KNNImputer, $k = 5$); two iterative imputers (sklearn.impute.IterativeImputer) based respectively on Bayesian ridge linear regression (sklearn.linear_model.BayesianRidge) and Random Forest regression (sklearn.ensemble.RandomForestRegressor). Once fitted any of these imputers, we have generated twelve $NxM$ matrices of probability estimates for each test set by sampling from a uniform distribution in the range [0,1], where N and M are, respectively, the numbers of variants (rows) and features (columns), and multiplying its values by one of the twelve values in the range 0.25–3.00, by a step of 0.25. Therefore, we added a missing value whenever any resulting number lower than the a priori probability of a missing value (precalculated for each feature and made available in Supplementary Data 9) appeared in the same position of the matrix. The metric we used to identify the best imputer for this study was the NRMSE defined as $E[X_{\text{true}} - X_{\text{imp}}]^2 / V[X_{\text{true}}]$, where $E$ and $V$ are, respectively, the expected value and the variance, $X_{\text{true}}$ is the true value, and $X_{\text{imp}}$ is the imputed value.

## Grid-Search cross-validation

We performed an exhaustive search over the parameters (or *hyperparameters*) of the tested ML methods using a stratified 10-fold Grid Search CV applied to a pre-computed and method-specific parameter grid (*GridSearchCV* module). The parameter grids for all methods are available in Supplementary Data 3. The preprocessed training set was further split into a 9-fold training set and a 1-fold test set. The Grid Search procedure was run for each CV iteration, for a total of $gs = 10 \times d$ times, where $d$ is the number of hyperparameter combinations explored. An ML method was trained and then tested on the 1-fold test set for each combination (Fig. 5d). The area under the precision-recall curve (auPRC) and under the receiver operating characteristic (auROC) curve metrics were evaluated to assess performance.

## Model training and testing

Each classifier was trained on the 19-fold training set using the best hyperparameter combination found by the Grid-Search loop and then tested on the relative 1-fold test set, which was not involved in the training or tuning phases (Fig. 5e). The best classifier exhibited a superior average auROC over the five replicas of the overall 20-fold CV process. Then, the total number of iterations performed for each ML method was $100 \times (gs + 1)$.

Finally, we trained a model of the best method on the entire Dataset 1, using the best hyperparameters found through the Grid-Search CV procedure, and scored all possible non-synonymous mtDNA variants.

## ML classifiers

The workflow described above was run with several ML classifiers. Most of them were implemented in the *scikit-learn* Python library.

**Support vector machine (SVM) with radial basis functions (RBF) kernel.** We implemented the *SVC module* of *scikit-learn*. It attempts to maximize the distance between two groups by minimizing the hinge loss function. The loss function was kept balanced by adjusting the class weights depending on their frequencies in the training set. The method was set to use a Radial Basis Function (*rbf*) as a kernel and L2 as the regularization term. The *rbf* function, defined as $K(x_1, x_2) = e^{(-\gamma ||x_1 - x_2||^2)}$, depends on a parameter γ, which is the inverse of the standard deviation of the *rbf* kernel and represents the influence of every single sample in the learning process, while L2 is controlled by a parameter $C$, which is inversely proportional to the regularization strength. Both parameters were optimized by Grid-Search.

**Gaussian Naive-Bayes (GNB) Balanced Bagging.** This is a probabilistic classifier that uses a Gaussian Naive-Bayes (GNB) model as the base learner for a Bootstrap Aggregation (Bagging) classifier. This approach is typically used to improve the performance of a "weak-learner" (such as the GNB) and reduce overfitting. To tackle the problem of class imbalance in our training set, we resorted to the Balance Bagging classifier implemented by the *imbalanced-learn* package of *scikit-learn*. We have also bootstrapped the features (without replacement) for each base-learner training in an attempt to reduce overfitting. The Grid-Search-based optimization step was applied to (i) the ratio of the bootstrapped samples over the total samples; (ii) the ratio of the bootstrapped features over the total number of features, and (iii) the GNB variance smoothing (more details in the *scikit-learn* documentation).

**K-Nearest Neighbors (KNN) Balanced Bagging.** 200 KNN classifiers were used as base learners for a Balanced Bagging classifier with the aim of reducing the risk of overfitting and balancing the pathogenic and neutral classes during the training phase. As in the previous case, the Grid-Search optimization step was used to find the (i) ratio of the bootstrapped samples over the total samples and (ii) the ratio of the bootstrapped features over the total number of features. Here,

another 2 hyperparameters were subjected to optimization: the number of "neighbors" to consider and the "weight" of each neighbor, which can be considered uniform or the inverse of their distances.

**Balanced Random Forest (RF).** This variation of the classical RF classifier is available from *imbalanced-learn* as the *BalancedRandomForestClassifier* module. It is an ensemble method where each tree in a forest will be provided with a balanced bootstrap sample for tree learning. Grid-Search was used to tune (i) the trees' maximum depth, (ii) the number of random features to consider for each node split, (iii) the minimum number of samples required to split a node, and (iv) the minimum number of samples that a node requires to be considered a leaf node.

**K-Nearest Neighbors (KNN) Bagging balanced through RUS and SMOTE.** We used a synthetic minority over-sampling technique available from the *imbalanced-learn* package called SMOTE[42] to solve the classification imbalance by generating synthetic samples in the pathogenic class of variants. We combined SMOTE with the randomly under-sampling technique (RUS) according to the following four steps: (i) training samples are bootstrapped with replacement, preserving the original training-set size; (ii) RUS undersamples the majority class (neutral variants) to partially decrease the imbalance; (iii) then, SMOTE is used to generate synthetic minority samples to perfectly balance the base-learner training-set; (iv) finally, the base-learner is fitted. Following this procedure, we implemented KNN_RusSmote, which uses a KNN classifier as a base learner. We chose to consider five neighbors in the SMOTE algorithm. As for KNN Balanced Bagging, Grid-Search was used to tune (i) the number of "neighbors" to consider, (ii) the "weight" of each neighbor, (iii) the ratio of the bootstrapped features over the total number of features, (iv) the ratio of the minority class over the majority class of samples after the RUS step.

**Scores and pathogenicity probabilities**
APOGEE 2 scores are decimal values ranging from 0 (benign) to 1 (pathogenic). Extreme values suggest a high level of confidence in the predictions. Scores were computed using the KNN_RusSmote machine learning algorithm, tuned through GS CV, and trained on the whole Dataset 1. Given that the algorithm is essentially a bagging model, we were able to determine the out-of-bag (OOB) APOGEE 2 score for each variant in the training set. We used the OOB score to infer the conditional APOGEE 2 probability distribution given the pathogenic class ($C = 1$); the APOGEE 2 probability distribution given the benign class ($C = 0$), instead, was inferred on the Dataset 4 (ClinGen's benign variants) scores, considering the OOB scores whenever they overlapped with Dataset 1 (training set). The distributions of both classes were approximated by Beta distributions, whose parameters were optimized through the SciPy Python library by minimizing the negative log-likelihood probability. Once inferred the distribution $P(y \mid C = 0)$ and $P(y \mid C = 1)$, where $y$ is the APOGEE 2 score, we could compute the posterior probability $P(C = 1 \mid y)$ using the Bayes theorem: $P(C = 1 \mid y) = P(y \mid C = 1)P(C = 1)/P(y)$, where $P(y)$ can be written in the form $P(y) = P(y \mid C = 1)P(C = 1) + P(y \mid C = 0)P(C = 0)$. We set the prior probability $P(C = 1) = 0.1$ as in[18].

**Misclassification and specificity assessment**
In order to calculate the per-class misclassification rate on Dataset 1, we first divided the APOGEE 2 scores into two classes (pathogenic and benign) based on the median value. Then, we calculated the frequency of incorrectly classified test variants, which are the variants that were disregarded from the model learning and tuning during the CV. Based on the distribution of APOGEE 2 scores for all MitImpact variants, the median threshold was recalculated for each test. Using MitImpact's non-benign frequency as the expected frequency for the null hypothesis, we performed a Chi-square test on the alternative hypothesis of

having a high frequency of benign variants in Datasets 2 and 3. Finally, we compared the average APOGEE 2 scores of Dataset 4's benign variants with the average APOGEE 2 scores of Dataset 1's benign variants, using a Mann–Whitney *U* test; the OOB scores were used for Dataset 1's benign variants in this test.

**The APOGEE 2's time-dependent learning curve**
From 2008 through 2020, we fetched the MITOMAP content every two years and generated seven datasets by applying the same criteria we used for Dataset 1 (Supplementary Data 5). We trained APOGEE 2 with these datasets and used the current MITOMAP version (July 2022) for testing the trained models. In particular, we applied a random stratified 5-fold partitioning strategy to the 2022 dataset, obtaining 5 partitions. For each partition, we trained a model on every other training set (2008–2020) and tested it on the partition, making sure to remove each variant already present in the partition from the training sets. The final auROC profile resulted from the average of the auROCs computed on the five different test partitions. The ML classifier used was still KNN_RusSmote; 10-fold Grid-Search CV has been performed on each training set.

**Positionality and impact of variants on the protein structure**
**Spatial autocorrelation of the APOGEE 2 scores.** Moran's index was used to measure the spatial autocorrelation of the predicted APOGEE 2 scores. It is defined as

$$I = \frac{N}{\sum_{i=1}^{N}\sum_{j=1}^{N}w_{ij}} \frac{\sum_{i=1}^{N}\sum_{j=1}^{N}w_{ij}(y_i - \bar{y})(y_j - \bar{y})}{\sum_{i=1}^{N}(y_i - \bar{y})^2}, \quad (1)$$

where $N$ is the number of spatial data points and $w_{ij}$ a weight matrix that measures the closeness of each residue in a pair. It ranges from −1 to +1; values significantly lower or higher than the expected value (i.e., $E[I] = \frac{-1}{N-1}$) indicate respectively negative or positive spatial autocorrelation[43]. We used a quadratic distance decay function, i.e., $w_{ij} = d_{ij}^{-2}$, to model the residue closeness in order to prioritize short-range autocorrelation.

Calculations were performed separately for each complex, while the APOGEE 2 scores were averaged by residue to have one value for every spatial point. Positional biases were limited by dropping all bagging base estimators, which were trained using at least one spatial feature (i.e., *X*, *Y*, or *Z* coordinates) from the APOGEE 2 predictions. 84 out of 200 base estimators were selected. This modification required recalculating the APOGEE 2 threshold between benign and pathogenic variants for this analysis. Due to the possibility that the APOGEE 2 scores obtained in this manner had a different distribution than the original one, we determined the pathogenicity threshold for this new distribution using the same formula we described in the "Model training and testing" methods section.

The weight matrix was row-normalized and then renormalized to ensure that $\sum_{i=1}^{N}\sum_{j=1}^{N}w_{ij} = 1$; the variable of interest, $y$, i.e., the APOGEE 2 average scores, was also standardized. Consequently $I = Z'WZ$, where $Z$ is the vector of the standardized values of $y$ and $W$ is the normalized weight matrix. *p*-values were calculated by permuting the APOGEE 2 scores 10,000 times and computing Moran's index for each permutation.

Since we were also interested in estimating the autocorrelation effect under a certain cutoff distance $r$, we performed the same procedure using a different weight matrix, defined as

$$w_{ij} = \begin{cases} 1 - \varepsilon, & d_{ij} \leq r \\ 0 + \varepsilon, & d_{ij} > r, \end{cases} \quad (2)$$

where $\varepsilon$ is an infinitesimal number ($10^{-6}$) whose purpose is only to make the weight matrix row-normalizable. We computed Moran's

index using this weight function at multiple cutoff distances, each time estimating a p-value on 1000 permutations.

Finally, we implemented a local version of Moran's index, LISA[44] or Local Indicators of Spatial Association, to investigate which regions contribute more to the global spatial autocorrelation. It is defined as $l = [l_1,...,l_N] = Z \cdot WZ$, where the weight matrix $W$ was introduced above and $\sum_{i=1}^{N} l_i = I$. High LISA values are assigned to residues contributing the most to the global positive spatial autocorrelation and showing similar APOGEE 2 scores to their neighbors. To confer significance to the results, we permuted the APOGEE 2 scores 1000 times and computed the $l$ vector for each permutation round; then, we calculated a p-value based only on the $l_i$ random distribution of the spatial point $i$. The False Discovery Rate was controlled using the Benjamini–Hochberg procedure.

**Molecular dynamics simulation.** Molecular dynamics (MD) simulations were conducted on the human respiratory complex I transmembrane arm (PDB ID: 5xtc). The starting system was refined using MODELLER ver. 9.16[45] and then mutated in silico using *UCSF Chimera* ver. 1.16[46] to introduce Ser34Pro, Ser34Tyr, Ser34Phe, and Thr35Pro amino acid variants in MT-ND3 and Ala72Val, Ala72Pro, and Phe46Leu variants in MT-ND6. Using the Membrane Builder Input Generator of the *CHARMM-GUI* web toolkit[47], the seven resulting alternative protein structures were embedded in a lipid bilayer composed of Palmitoyl-2-oleoyl-sn-glycero-3-phosphocholine (POPC), 1-palmitoyl-2-oleoyl-sn-glycero-3-phosphoethanolamine (POPE), and 1,1',2,2'-tetraoleoyl-cardiolipin (TOCL) to mimic the composition of the inner mitochondrial membrane that forms tight interactions with the transmembrane helix of each protein. Then, both the proteins and the lipid bilayer were solvated in a periodic boundary condition box filled with 22.5 Å of TIP3P water molecules on either side of the bilayer, and a salt concentration of 0.15 M KCl was added to obtain a neutral simulation system.

All generated systems were energy minimized and equilibrated following the CHARMM-GUI's workflow, consisting of (i) steepest-descent minimization for 5000 steps; (ii) a gradual heating process conducted into a canonical ensemble (NVT) for 250,000 steps with a timestep of 1 fs; (iii) an isothermal-isobaric ensemble equilibration for 125,000 steps with a timestep of 1 fs followed by 750,000 steps of an isothermal-isobaric ensemble equilibration every 2 fs. During the equilibration, harmonic restraints were applied to both the heavy protein atoms and the membrane lipid phosphates.

Gaussian accelerated MD (GaMD)[48] simulations were performed, starting with a 20 ns short classical MD simulation used to collect potential statistics for calculating the GaMD acceleration parameter, followed by a ~80 ns equilibration run. Finally, 200 ns of GaMD simulations were carried out, divided into 10 sequential production steps. Three replicas of the simulation of the wild-type and each mutant system were performed using *Amber20*[49]. Every GaMD simulation was performed at the "dual-boost" level, with one boost applied to the total potential energetic term and the other to the dihedral energetic term. For both the dihedral and the total potential energetic terms, 12.0 kcal/mol was chosen as the upper limit of the boost potential SD, $\sigma_0$. The code for the entire simulation workflow is provided in Supplementary Software 1.

**Analysis of simulated trajectories.** *AmberTools21* was used to calculate the Root-Mean-Square Fluctuation (RMSF), which measures the time deviation of the positions of the atomic coordinates of the alpha carbons of each residue and those of the reference starting structure. The *GetContacts* (https://getcontacts.github.io) tool was used to compute all the atomic interactions and contacts established in each time frame of the simulated trajectories. Secondary structures were computed using the DSSP module of the MDTraj[50] tool for each simulation frame. The DSSP assignments were calculated using the

8-category schemes: H = α helix, B = residue in isolated beta-bridge, E = extended strand, participating in the beta ladder, G = 3-helix (3/10 helix), I = 5-helix (π helix), T = hydrogen bonded turn, S = bend, – = unclassified. 3D images and motions were generated using UCSF Chimera.

## Reporting summary

Further information on research design is available in the Nature Portfolio Reporting Summary linked to this article.

## Data availability

The datasets supporting the conclusions of this article are included within the article and its Supplementary Data files. The APOGEE 2 probabilities/classes of pathogenicity can be freely downloaded from MitImpact [http://mitimpact.css-mendel.it]. Datasets 2 and 3 are freely available from gnomAD [https://gnomad.broadinstitute.org] and HelixMTdb [https://www.helix.com/pages/mitochondrial-variant-database]. Dataset 4 is freely available from MITOMAP [https://mitomap.org/MITOMAP/Benign]. ClinVar is freely available at [https://ftp.ncbi.nlm.nih.gov/pub/clinvar/].

## Code availability

The complete molecular dynamics simulation protocol is available as Supplementary Software 1. The APOGEE 2 machine-learning workflow is available from https://github.com/mazzalab/playgrounds[51] and is freely runnable as a Colab notebook.

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

## Acknowledgements

This research was funded by the Italian Ministry of Health (Ricerca Corrente 2022–2024) and the "5 × 1000" voluntary contributions and supported by NVIDIA Corporation and the Amber team (TM). Additional support came from National Institutes of Health grants NS021328, MH108592, OD010944, and CA259635 (D.C.W.).

## Author contributions

S.D.B. designed, implemented, and benchmarked APOGEE 2 with the help of L.P.; F.P. and T.B. performed molecular dynamics simulations; A.G. contributed with data curation; N.L. and A.N. validated APOGEE 2 and helped make it available as a public notebook; S.Z. and M.L. provided temporal MITOMAP data; M.L. and V.P. contributed to the definition of variant pathogenicity criteria; T.M. conceived the project; T.M. and V.C. supervised, respectively, the computational and genetic aspects of the work with the assistance of M.C., A.L.V., and D.C.W. T.M. wrote the paper, with contributions from all authors.

## Competing interests

The authors declare no competing interests.
