## [Peer Review File · Nature Communications]

APOGEE 2: multi-layer machine-learning model for the interpretable prediction of mitochondrial missense variantsREVIEWER COMMENTS

Reviewer #1 (Remarks to the Author):

This paper presents data that will be useful and accessible to the mtDNA community via the most frequently access portal or such information.

The abstract could have a greater impact; currently, it lacks impact starting with lots of basic mitochondrial/mtDNA science. Beginning the Abstract start at 'APOGEE2, a mitochondrial-centred ensemble method....' highlighting the benefit of re-working this tool in the mitochondrial context would attract more people to read the article.

? should the title not say APOGEE2 rather than APOGEE

The work here presents a newer version of an already published method with the original work published in PLoS Biology 2017.

Note that the original work also delivered a list of scores for all 24,189 possible non-synonymous amino acid changes potentially affecting the human mitochondrial DNA. In the original paper, the 2017 authors provide a detailed description of the algorithm. Some supporting files here provide similar details; can the differences between the algorithms be more clearly highlighted? This work combines the computational expertise of these who published the tool with eminent researchers in the mitochondrial field.

One of the paper's main thrusts is to re-calculate all possible missense variants of the mtDNA.

'We have pre-calculated the prediction scores and pathogenicity probabilities of all possible missense variants of the mtDNA and set five classes of pathogenicity to support clinicians and geneticists in reporting their genetic diagnoses'

Briefly, in the context of the re-calculation using the APOGEE2 program. Have the authors considered how their scores compare to the calculated scores for all possible missense variants of the mtDNA using two methods (MutPred/ mtDNA selection score) published by Luisa Pereira et al in the AJHG (2009)

Figure 2: looks at a number of prediction software tools but not MutPred this has been used frequently in the mitochondrial context including by the pipeline Hansi Weißensteiner. Why not MutPred?

The paper also considers mechanistic analysis of specific amino acids 120 that cause perceptible long-range effects on the protein structure. In terms of this prediction only a brief mention made of the importance of haplogroup context. This is an area of challenge for all such predictions and has been a topic of recent debate, but this is not covered in-depth here. The paper would be more substantial with a little more emphasis here as it will also affect the mechanistic analysis.

The list of variants made available as a CVS file will be useful to the field.

Reviewer #2 (Remarks to the Author):

Salvatore and colleagues present here a second version of APOGEE, a machine learning based pathogenicity predictor designed for variants in the mitochondrial genome. First APOGEE algorithm seemed to be very accepted and this second version overtakes its own performance as well as the rest of the predictors. I am pretty sure that next step in the pathogenicity prediction field is to go specific, build good training sets on more specific datasets than genome wide and tune them there in order to gain predicting power in the real application. So I believe the authors are doing very good in this sense and their experience can indeed guide other initiatives. Said that, I have some questions, suggestions and concerns that the authors might want to address, see below.

In the introduction I miss a clear statement where there is a reference to APOGEE as an existing algorithm. I think the reader would get more context if this is done.

It is strange to me to find Figure 1 describing data commented in the introduction, also Supplementary Figures embedded in the main text, however I guess it is not my task to evaluate if this is feasible, and probably formatting is not authors responsibility at this stage neither.

Regarding the training set, which is the criterium to classify the pathogenicity of the variants in the training set? Please comment limitations in the discussion section.

Line 127: I would suggest that in the results, authors do not make any judgment on the level of the correlation between features, adjectives as strong or weak are better avoided. Same in line 133 with the expression "moderately low values".

Please be more informative on the Supplementary Figure 1. The reader would not understand what is being compared and what is the comparison for. Likewise, please highlight the names of the features that are not pathogenicity predictors.

I wonder why CADD does not behave well as it is itself a machine learning methods trained over the same features as APOGEE, or some of them at least. So, can you comment on which features overlap between the two methods?

Line 134: I guess the details on the imputation method is better fitted within Materials and Methods.

It is interesting the comments on the distribution of the APOGEE score. In principle, a bimodal distribution would be a good predictor as two peaks would correspond to benign and pathogenic. In fact, much better than the probability density distribution that the authors believe others fit to, is that right? So, is there any other predictor behaving similarly? If I had to bet I would say all, and did APOGEE v1 had a similar performance?

Figure 3E is mentioned before Figure 3B, C, D, please sort them accordingly. Using the same colors for Figure 3A and 3B can make wrong conclusions, the capture does not help much. In a first view I would say both represent APOGEE scores.

In the results I do not get where the 24190 variants come from, are they from gnomAD? A mito resource? An internal dataset? Or by “all possible” do they mean all the possible combination for missenses? In the section, then the authors start giving results for Dataset 1, 2 and 3. The first one is the training dataset, but the others I don’t know to what they correspond. Are they blind datasets or training ones? Please explain somewhere they are named.

In the results section “Variant functionality prediction” the authors should mention they tested only in mito variants, otherwise it seems they applied APOGEE to all variants (“As a further test of the negative variants, we used APOGEE to score every missense variant in the gnomAD v3.1.2 and HelixMTdb databases”).

Dataset 4 is also named afterwards, did I miss anything?

I would like to know which is the criteria to set up the threshold for the variants’ clinical classification (benign, likely benign, vus, likely pathogenic, and pathogenic). The range for VUS is 0.1 to 0.9 which is the wider assigned. Which proportion of VUS do they obtain? And it is lower or greater than the proportion found in ClinVar (or any other resource) in the nuclear and the mito genome?

Since VUS are a huge bottleneck in clinical evaluation of genomic variants, could the authors work within this subset to fine-tune predictions just there? For instance, what they would think on using as training set the VUS reclassified over time in any of the mito resources (or ClinVar).

What exactly is the authors proposal in the use of molecular dynamics in the evaluation of the mito variants? Are they going to introduce these algorithms into the evaluation somehow? Are they planning to detect somehow the variants that have non-local effects?

Reviewer #3 (Remarks to the Author):

In this study, the authors updated the APOGEE predictor for prioritizing potential pathogenic missense variants in mitochondrial DNA. Using a series of testing datasets, they benchmarked their approach, which outperformed all competitors among most of the testing datasets. The manuscript is generally well-written, and based on its results, this update over the original method is substantial. To improve the clarity of the manuscript, there are some points that I’d like the authors to address:

1. I would give datasets 1, 2, 3 etc., a meaningful name representing their biological differences. Also, it is not clear why these specific 4 constructions of datasets were used. It seems that specificity is given more attention than sensitivity given 3 out of 4 datasets were used to evaluate specificity.

2. What does it mean by “MITOMAP confirmed” variants? The occasional referring to variants as “confirmed” can be confusing (line 81). What are the differences between “confirmed” and “reported” variants?

3. It can be confusing when the term APOEGEE was used to refer APOEGEE2 (line 303).

4. The conclusion of APOGEE’s ability to “quantify the level of certainty” mentioned in line 282 was not well explained. Are ClinGen’s neutral variants considered to have a lower pathogenicity probability than those used in Dataset 1? It is better to give a brief overview of the meaning of the 4 datasets before diving into the detailed discussion of the results. Additionally, if those ClinGen’s neutral variants are more neutral, it may be better to use them as training data to reduce the noise in the training process.

5. As you stated under the section “Evaluation of variants that alter the protein structure non-locally”, some variants that alter distal protein structure can be difficult to predict by APOEGEE. Did you mean APOEGEE2 in this case? If so, how do you think the new structural features contributed to these incorrect predictions?

Reviewer #4 (Remarks to the Author):

This work is about developing a prediction method for interpreting mitochondrial missense variants, Apogee. The same group published a prior version of Apogee 2017, and this new method is its upgraded one, version 2. When I read their previous paper (Castellana S, Fusilli C, Mazzoccoli G, Biagini T, Capocefalo D, Carella M, et al. (2017) High-confidence assessment of functional impact of human mitochondrial non-synonymous genome variations by APOGEE. PLoS Comput Biol 13(6): e1005628) and compared it to this current work, I didn't find any significant differences between the two versions. It seems like that Apogee 2 is essentially the same program with some upgraded performance and capabilities, mainly due to the graded dataset.

REVIEWER COMMENTS

Reviewer #1:

This paper presents data that will be useful and accessible to the mtDNA community via the most frequently access portal or such information.

The abstract could have a greater impact; currently, it lacks impact starting with lots of basic mitochondrial/mtDNA science. Beginning the Abstract start at 'APOGEE2, a mitochondrial-centred ensemble method....' highlighting the benefit of re-working this tool in the mitochondrial context would attract more people to read the article.

? should the title not say APOGEE2 rather than APOGEE

> We have added 'APOGEE 2' in the title and reworded the abstract to make it as informative and attractive as possible.

The work here presents a newer version of an already published method with the original work published in PLoS Biology 2017.

Note that the original work also delivered a list of scores for all 24,189 possible non-synonymous amino acid changes potentially affecting the human mitochondrial DNA. In the original paper, the 2017 authors provide a detailed description of the algorithm. Some supporting files here provide similar details; can the differences between the algorithms be more clearly highlighted? This work combines the computational expertise of these who published the tool with eminent researchers in the mitochondrial field.

> APOGEE version 2 was completely revamped, and a detailed description of the differences between the two versions was summarized for reviewer 4 below and reported in the manuscript.

One of the paper's main thrusts is to re-calculate all possible missense variants of the mtDNA. 'We have pre-calculated the prediction scores and pathogenicity probabilities of all possible missense variants of the mtDNA and set five classes of pathogenicity to support clinicians and geneticists in reporting their genetic diagnoses'

Briefly, in the context of the re-calculation using the APOGEE2 program. Have the authors considered how their scores compare to the calculated scores for all possible missense variants of the mtDNA using two methods (MutPred/ mtDNA selection score) published by Luisa Pereira et al in the AJHG (2009)

Figure 2: looks at a number of prediction software tools but not MutPred this has been used frequently in the mitochondrial context including by the pipeline Hansi Weißensteiner. Why not MutPred?

> We have measured MutPred's (first and newer versions) performance (auROC and auPRC) and compared it with that of APOGEE 2. We have added this information as a new Supplementary Figure 3 and commented on it in this revised version. MutPred was not among the best performing tools (see figure below). For this reason, we did not include it in our training workflow. However, we have taken this opportunity to show additional comparisons between APOGEE 2 and other similar tools: EFIN, MutationAssessor, and CADD, which, in turn, were used during APOGEE 2's training step. APOGEE 2 outperforms all of them.

The paper also considers mechanistic analysis of specific amino acids 120 that cause perceptible long-range effects on the protein structure. In terms of this prediction only a brief mention made of the importance of haplogroup context. This is an area of challenge for all such predictions and has been a topic of recent debate, but this is not covered in-depth here. The paper would be more substantial with a little more emphasis here as it will also affect the mechanistic analysis.

> We agree with the reviewer that the haplogroup context represents an important aspect that should be considered to predict a functional effect on variants. The examples we reported in this work address the issue of long-range effect, and we underline the challenging task of predicting haplogroup variants as they can be common in control populations and involve poorly conserved residues. Moreover, we also discuss how this can impact the interpretation of synergistic variants.

As mentioned in the discussion, performance improvements are possible by adding more accurate structural features to variants and accounting for their epistatic interactions. Based on this, a pilot project of molecular dynamics simulations of a significantly high number of variants that may distantly alter the complex structure or cooperate with other variants to cause a significant respiratory-chain deficiency has already started.

The list of variants made available as a CVS file will be useful to the field.

> The complete list of variants together with the APOGEE 2 scores and several other annotations have been made freely available from our MitImpact database (top right, "Download" dropdown menu) at the URL: https://mitimpact.css-mendel.it/cdn/MitImpact_db_3.1.0.txt.zip

Reviewer #2:

Salvatore and colleagues present here a second version of APOGEE, a machine learning based pathogenicity predictor designed for variants in the mitochondrial genome. First APOGEE algorithm seemed to be very accepted and this second version overtakes its own performance as well as the rest of the predictors. I am pretty sure that next step in the pathogenicity prediction field is to go specific, build good training sets on more specific datasets than genome wide and tune them there in order to gain predicting power in the real application. So I believe the authors are doing very good in this sense and their experience can indeed guide other initiatives.

> Thank you

Said that, I have some questions, suggestions and concerns that the authors might want to address, see below.

In the introduction I miss a clear statement where there is a reference to APOGEE as an existing algorithm. I think the reader would get more context if this is done.

> We have anticipated the bibliographic reference of APOGEE (version 1) already in the introduction and specified the versions: “1” for the former, already published, and “2” for the current version. We hope this clarifies.

It is strange to me to find Figure 1 describing data commented in the introduction, also Supplementary Figures embedded in the main text, however I guess it is not my task to evaluate if this is feasible, and probably formatting is not authors responsibility at this stage neither.

> During submission, we encountered this note: “We are flexible with regard to the format of initial submissions but to facilitate the review process we strongly encourage you to incorporate the manuscript text and figures into a single pdf or Microsoft Word file up to 30 MB in size. Figures may be inserted within the text at appropriate positions or grouped at the end. Each figure legend should be presented on the same page as its figure.”

To facilitate the reviewers' job, we therefore decided to do the same for Supplementary Figures, sure that the manuscript would be completely reformatted in case it was accepted for publication.

Regarding the training set, which is the criterium to classify the pathogenicity of the variants in the training set? Please comment limitations in the discussion section.

> The variants considered pathogenic in the training set (Dataset 1) come from MITOMAP. In the introduction, we have defined, and slightly better explained in this revision, “confirmed” and “reported” MITOMAP variants as those with compelling scientific evidence and those that have not been definitively shown to cause disease, respectively (rows 64-67). We have extensively discussed their distribution along protein coding mitochondrial genes, as also shown in Figure 1. Then, in Methods > Datasets (rows 419-420), it is clearly explained that “The pathogenic set was made up of MITOMAP “disease” variants, including 41 confirmed and 99 reported variants.”

Unfortunately, Nat Comm required us to place Methods after Results and Discussion. This may have caused some confusion. For this reason, we have referred to Methods for further clarifications at line 126.

Line 127: I would suggest that in the results, authors do not make any judgment on the level of the correlation between features, adjectives as strong or weak are better avoided. Same in line 133 with the expression “moderately low values”.

> Amended

Please be more informative on the Supplementary Figure 1. The reader would not understand what is being compared and what is the comparison for. Likewise, please highlight the names of the features that are not pathogenicity predictors.

> Supplementary Figure 1A is a matrix of correlation values between all pairs of features. In particular, each Pearson correlation coefficient represented in the matrix (i.e., each cell) comes from the prediction scores of all mitochondrial missense variants, scaled to zero mean and unit variance, calculated by all pairs of non-positional predictors considered in this work (namely, those not directly involving the genomic or amino acidic position of the variant, as we retain that these features were not directly relatable to variant pathogenicity). This clarification was added in the method section. Supplementary Figure 1B's meaning has been explained in the method section. The figure legend was improved accordingly.

I wonder why CADD does not behave well as it is itself a machine learning methods trained over the same features as APOGEE, or some of them at least. So, can you comment on which features overlap between the two methods?

> Reasons for their different performances can be attributed to (i) the training set: nuclear variants for CADD and mitochondrial variants for APOGEE 2; (ii) the machine learning algorithm: a support vector machine with a linear kernel for CADD and KNN Bagging balanced through Random Under-Sampling (RUS) and Synthetic Minority Oversampling Technique (SMOTE) for APOGEE 2. Thus, while the CADD scoring system shares some features with APOGEE2: vertebrate PhastCons, vertebrate PhyloP, SIFT, and Polyphen 2, as the reviewer stated, APOGEE 2 exploits mitochondrial specific features as well, such as the protein global free energy upon mutation ($\Delta\Delta G$), an empirical mitochondrial amino acid substitution model (MtMam), mitochondrial-specific phylogenetic conservation scores, genomic and spatial coordinates of nucleotides and amino acids, and variant deleteriousness predictions taken from several scoring systems (e.g. MutationAssessor and EFIN). (iii) CADD is a predictor, whereas APOGEE 2 is a meta-predictor and aggregates the predictions of other predictors as well. In version 2, we attempted to include CADD, but it performed badly and, therefore, was dropped by our feature importance analysis (see Figure 2B on feature importance).

Line 134: I guess the details on the imputation method is better fitted within Materials and Methods.

> We have removed the text “with 100 decision tree regressors”, which was moved to Methods, and left the mention of the imputer and the fact that it was chosen because it performed better than all others we have tested.

It is interesting the comments on the distribution of the APOGEE score. In principle, a bimodal distribution would be a good predictor as two peaks would correspond to benign and pathogenic. In fact, much better than the probability density distribution that the authors believe others fit to, is that right? So, is there any other predictor behaving similarly? If I had to bet I would say all, and did APOGEE v1 had a similar performance?

> In Figure 3A, we reported the frequency distribution of the APOGEE 2 scores for all the possible missense mtDNA variants. We agree with the reviewer that the distribution obtained resembles a bimodal distribution, and we have no doubt that several other tools have a similar distribution, e.g., the recently developed MetaRNN, <https://doi.org/10.1186/s13073-022-01120-z>. In this tool, they have in fact set the cutoff to 0.5, as we did with APOGEE 1, which equally split the two curves of the distribution.

With APOGEE 2, we have chosen instead to adhere to the American College of Medical Genetics and Genomics/AMP guidelines (references [11] and [17]) and, hence, to calculate a probability density function and split variants into five classes according to their pathogenicity probabilities.

Figure 3E is mentioned before Figure 3B, C, D, please sort them accordingly.

> Amended

Using the same colors for Figure 3A and 3B can make wrong conclusions, the capture does not help much. In a first view I would say both represent APOGEE scores.

> Yes, the colors of both subfigures have the same meaning. They represent APOGEE 2 scores. This was made clear in the figure legend. Thanks.

In the results I do not get where the 24190 variants come from, are they from gnomAD? A mito resource? An internal dataset? Or by “all possible” do they mean all the possible combination for missenses?

> Yes, 24190 is the number of all possible missense mitochondrial nucleotide variants, existing or currently unknown. We have precomputed them and made them available from both our MitImpact database and Supplementary Table 5 in this work.

In the section, then the authors start giving results for Dataset 1, 2 and 3. The first one is the training dataset, but the others I don't know to what they correspond. Are they blind datasets or training ones? Please explain somewhere they are named.

> We have provided a detailed description of the four datasets in the first chapter of the Methods section. The chapter was entitled “Datasets”.

In the results section “Variant functionality prediction” the authors should mention they tested only in mito variants, otherwise it seems they applied APOGEE to all variants (“As a further test of the negative variants, we used APOGEE to score every missense variant in the gnomAD v3.1.2 and HelixMTdb databases”).

> Amended, thank you.

Dataset 4 is also named afterwards, did I miss anything?

> As anticipated above, we have provided a chapter (“Datasets” in Methods) that describes where the datasets come from and how they were obtained.

I would like to know which is the criteria to set up the threshold for the variants’ clinical classification (benign, likely benign, vus, likely pathogenic, and pathogenic). The range for VUS is 0.1 to 0.9 which is the wider assigned. Which proportion of VUS do they obtain? And it is lower or greater than the proportion found in ClinVar (or any other resource) in the nuclear and the mito genome?

> We adopted the pathogenicity thresholds recommended by the American College of Medical Genetics and Genomics/AMP (references [11] and [17]). Therefore, we have calculated the APOGEE 2 probability density function and split all possible mitochondrial missense variants into five classes according to those thresholds. The probability range of VUS variants is the widest since it collects all variants that do not meet the ACMG/AMP pathogenicity criteria. In particular, “The ACMG/AMP defined the term “likely pathogenic” to mean >90% certainty of a variant being disease-causing, but below a higher “pathogenic” threshold. In Bayesian terms, these translate to a Post_P of >0.90 for likely pathogenic” [17]. At the same time, ACMG/AMP adopts an a-priori probability of pathogenicity, the Prior_P, which was set to 0.1. The conclusion is that if none of the pathogenicity criteria are met, a variant is a VUS, and thus the prior must be between 0.10 and 0.90, which is the Post_P range for a VUS.

The number of VUS predicted by APOGEE 2 and those found in a clinical database like ClinVar are not comparable, since APOGEE 2 predicts the pathogenic impact of all possible mitochondrial missense variants (including those never observed), while ClinVar reports only variants actually found by the scientific community. This is the first important bias. Moreover, the level of “curation” of the variants in the training set of APOGEE 2 and that of ClinVar is different. ClinVar is not as specialized in mitochondrial genetics as MITOMAP is. This often causes variants to swap pathogenicity assessments in ClinVar through releases. This is a second important bias. Hence, while APOGEE 2 classifies 44.33% of all mitochondrial missense variants as VUS and ClinVar (July 2022) does it for 42.07%, we do not feel confident to report this comparison in the main manuscript even if the difference between these two proportions is not that big, and this could advantage APOGEE 2's predicting skills.

Since VUS are a huge bottleneck in clinical evaluation of genomic variants, could the authors work within this subset to fine-tune predictions just there? For instance, what they would think on using as training set the VUS reclassified over time in any of the mito resources (or clinvar).

> We have provided a paragraph at the end of the discussion section that describes how we intend to narrow down the number of VUS variants in future releases of APOGEE.

What exactly is the authors proposal in the use of molecular dynamics in the evaluation of the mito variants? Are they going to introduce these algorithms into the evaluation somehow? Are they planning to detect somehow the variants that have non-local effects?

> A wide number of mitochondrial pathogenic mutations target the structural integrity of a functional domain and, unfortunately, this aspect could be misinterpreted when focusing on local

aspects and ignoring data about their structural long-range effects. For this reason, we introduced molecular dynamics to show how advanced structural approaches could support the interpretation of a variant and highlight putative disruptive mechanisms.

The data obtained from this pilot study were so insightful that they delineated a grand challenge consisting of the structural analysis of all mitochondrial missense variants to further integrate in a “structure-focused” Mitimpact. The next step in this direction would consist of massively parallel simulations of all known missense variants hitting the 13 proteins encoded by the mtDNA, starting from the ones mapping on mitochondrial Complex I and, thus, expanding to all possible amino acid changes in Complex I and, finally, to other complexes. This would allow us to characterize the key conformations and kinetic properties of all these variants and explore their long-range effects, thereby speculating on the role of putative co-occurring variants. This latter point is especially important for variants with divergent pathogenicity assessments, which are primarily brought on by the presence of one or more co-inherited missense variants in the same protein or structurally/functionally related proteins that affect the functional role of a mutant residue.

Reviewer #3:

In this study, the authors updated the APOGEE predictor for prioritizing potential pathogenic missense variants in mitochondrial DNA. Using a series of testing datasets, they benchmarked their approach, which outperformed all competitors among most of the testing datasets. The manuscript is generally well-written, and based on its results, this update over the original method is substantial.

> Thank you

To improve the clarity of the manuscript, there are some points that I'd like the authors to address:

1. I would give datasets 1, 2, 3 etc., a meaningful name representing their biological differences. Also, it is not clear why these specific 4 constructions of datasets were used. It seems that specificity is given more attention than sensitivity given 3 out of 4 datasets were used to evaluate specificity.

> Despite our efforts to find more informative labels, replacing 'Dataset 1' with acronyms such as 'Dataset TS' (training set) would likely yield little improvement in the reader's understanding. Alternatively, we could have used lengthier descriptors, but repeatedly referring to these datasets throughout the paper may have hindered readability. Therefore, we opted for a middle ground by directing the reader to the first chapter of the Methods section, where all four datasets are thoroughly described upon their initial mention in the Results section, which occurred as early as the second line (line 126).

2. What does it mean by "MITOMAP confirmed" variants? The occasional referring to variants as "confirmed" can be confusing (line 81). What are the differences between "confirmed" and "reported" variants?

> We have clearly defined what "confirmed" and "reported" variants are at rows 64-67 of the Introduction ("Mutations in the mtDNA are at the core of many human diseases. Currently, ~1000 different mutations are associated, based on literature, with human diseases in MITOMAP, ~10% of which with compelling evidence of being pathogenic based on published literature (named as confirmed in MITOMAP), ~86% not been definitively shown to cause disease (named as reported), and 4% including synergistic and conflicting variants."). Their definition comes from MITOMAP. In detail, "confirmed" status indicates that "at least two or more independent laboratories have published reports on the pathogenicity of a specific mutation. These mutations are generally accepted by the mitochondrial research community as being pathogenic". "Reported" status indicates only a possible condition of a pathologic effect based on publications. Therefore, this classification is based on published literature. To be clearer, we added the criteria to define a variant as confirmed/reported by MITOMAP the first time we cited them.

3. It can be confusing when the term APOEGEE was used to refer APOEGEE2 (line 303).

> We have specified the APOGEE version we refer to throughout the manuscript to avoid confusion.

4. The conclusion of APOGEE's ability to "quantify the level of certainty" mentioned in line 282 was not well explained. Are ClinGen's neutral variants considered to have a lower pathogenicity probability than those used in Dataset 1? It is better to give a brief overview of the meaning of the 4 datasets before diving into the detailed discussion of the results. Additionally, if those ClinGen's neutral variants are more neutral, it may be better to use them as training data to reduce the noise in the training process.

> We have better explained the fact that ClinGen's neutral variants are manually curated, therefore, more likely to be neutral than Datasets 2 and 3, which in turn contain supposedly neutral variants based on (less stringent) allelic frequency considerations alone. This is now clear in the Results section (lines 278-280) and in Methods (Datasets chapter).

Benchmarking specificity was our Achilles' heel, since the neutral subset of our Dataset 1 was "less curated" than the pathogenic counterpart. When we were made aware of the release of ClinGen's mtDNA variant assessments on September 7, 2022, APOGEE 2 was already trained, tested, and validated. We considered the possibility of redoing all our work, but instead opted to use ClinGen's neutral variants to create a high-quality validation set for APOGEE 2. This allowed us to test the prediction capabilities of APOGEE 2, where it was weaker. Alternatively, we could have trained APOGEE 2 on ClinGen's neutral variants, but then we would have been unable to properly validate our predictions.

5. As you stated under the section "Evaluation of variants that alter the protein structure non-locally", some variants that alter distal protein structure can be difficult to predict by APOEGEE. Did you mean APOEGEE2 in this case? If so, how do you think the new structural features contributed to these incorrect predictions?

> Yes, we meant APOGEE 2. We have specified the version.

Unfortunately, the structural features currently introduced in APOGEE 2 do not cover long-range effects. They provide positional information, which contributed to inferring tolerance regions for variants, and energetic regions, which contributed to assessing the consequence of a variant on the global structure of a protein. The detection of point-to-point long-range effects is our aim, as per our answer to the last comment of Reviewer #2.

Reviewer #4:

This work is about developing a prediction method for interpreting mitochondrial missense variants, Apogee. The same group published a prior version of Apogee 2017, and this new method is its upgraded one, version 2. When I read their previous paper (Castellana S, Fusilli C, Mazzoccoli G, Biagini T, Capocefalo D, Carella M, et al. (2017) High-confidence assessment of functional impact of human mitochondrial non-synonymous genome variations by APOGEE. PLoS Comput Biol 13(6): e1005628) and compared it to this current work, I didn't find any significant differences between the two versions. It seems like that Apogee 2 is essentially the same program with some upgraded performance and capabilities, mainly due to the graded dataset.

> Compared with the first and rudimentary release of APOGEE version 1, whose classification method was very simple and based on a training process carried out on a limited training set and without a validation set, what we present in this paper is a study of the "fragility" of the mitochondrial genome performed through our machine-learning-based method. In particular:

- We present a more sophisticated learning machine that takes into account the imbalance of the subclasses of the training set (i.e., neutral and pathogenetic variants). We have released an executable notebook to rebuild the predictor from start-to-finish from my group's GitHub. This research is completely repeatable and, as a side note, this could inspire many who embark on the definition of predictors similar to ours. The ground truth is composed of 4 manually curated datasets that we use to train and validate the results. This has never been done for the mitochondrial genome. We compare the predictability of our method with several alternative methods and declare the superiority of APOGEE 2 for all measured performance metrics.
- We generated and added some unique structural features to the learning machine; they are chemical, physical, electrical properties of mitochondrial protein residues, positional features (genomics and in the 3D structures of proteins), conservation in species.
- We identify fragile regions in mitochondrial protein structures, obtained through a spatial autocorrelation study. This has never been done.
- We identified amino acids that cause perceptible long-range effects on the protein structure through molecular dynamics simulation techniques and applied them on some exemplary cases that we have come across in our clinical practice. Here too we made the entire simulation workflow public to repeat this analysis. Then we also explained hidden molecular/structural mechanisms.
- Thanks to the collaboration with the authors of MITOMAP (reference database for the census of mitochondrial variants and their association with disease and our co-authors in this paper), we were able to access the historical data of MITOMAP and reconstruct the learning process of our method from 2008 to today and define future developments in order to improve its classification performance.
- We have pre-calculated and made available the predictions of all possible missense variants of the mitochondrial genome, i.e., 24190, through our portal <https://mitimpact.css-mendel.it/>. To make life easier for clinicians and geneticists in reporting their genetic diagnoses, we have defined 5 classes of pathogenicity through Bayesian reasoning.

REVIEWERS' COMMENTS

Reviewer #1 (Remarks to the Author):

Concerns have been addressed in the revisions -

I look forward to using the tool!

Reviewer #2 (Remarks to the Author):

Authors have addressed all my concerns and answer my questions/doubts quite reasonably. Thanks

Reviewer #3 (Remarks to the Author):

The authors have addressed all my concerns.

Reviewer #4 (Remarks to the Author):

After reading the previous manuscript, I was concerned that this new work was in essence a mere extension of Apogee from the same group, published 2017. In response, the authors summarized the contributions of this new work, which I found seemed reasonable, and made my judgement more positively. The change in my sentiment toward this paper was also affected by the comments raised by other reviewers and author's responses.